# Identification of a Novel Germline *PPP4R3A* Missense Mutation Asp409Asn on Familial Non-Medullary Thyroid Carcinoma

**DOI:** 10.3390/biomedicines12010244

**Published:** 2024-01-22

**Authors:** Yixuan Hu, Zhuojun Han, Honghao Guo, Ning Zhang, Na Shen, Yujia Jiang, Tao Huang

**Affiliations:** 1Department of Breast and Thyroid Surgery, Union Hospital, Tongji Medical College, Huazhong University of Science and Technology, Wuhan 430022, China; emery1030@163.com (Y.H.); u201510429@alumni.hust.edu.cn (Z.H.); guohonghao0518@163.com (H.G.); zhangning0823@126.com (N.Z.); nashen@hust.edu.cn (N.S.); 2Department of Dermatology, Traditional Chinese and Western Medicine Hospital of Wuhan, Tongji Medical College, Huazhong University of Science and Technology, Wuhan 430022, China

**Keywords:** familial papillary thyroid cancer, predisposition, germ-line mutations, PPP4R3A

## Abstract

Familial non-medullary thyroid carcinoma (FNMTC) accounts for 3% to 9% of all thyroid cancer cases, yet its genetic mechanisms remain unknown. Our study aimed to screen and identify novel susceptibility genes for FNMTC. Whole-exome sequencing (WES) was conducted on a confirmed FNMTC pedigree, comprising four affected individuals across two generations. Variants were filtered and analyzed using ExAC and 1000 Genomes Project, with candidate gene pathogenicity predicted using SIFT, PolyPhen, and MutationTaster. Validation was performed through Sanger sequencing in affected pedigree members and sporadic patients (TCGA database) as well as general population data (gnomAD database). Ultimately, we identified the mutant *PPP4R3A* (NC_000014.8:g.91942196C>T, or NM_001366432.2(NP_001353361.1):p.(Asp409Asn), based on GRCH37) as an FNMTC susceptibility gene. Subsequently, a series of functional experiments were conducted to investigate the impact of *PPP4R3A* and its Asp409Asn missense variant in thyroid cancer. Our findings demonstrated that wild-type *PPP4R3A* exerted tumor-suppressive effects via the Akt-mTOR-P70 S6K/4E-BP1 axis. However, overexpression of the *PPP4R3A* Asp409Asn mutant resulted in loss of tumor-suppressive function, ineffective inhibition of cell invasion, and even promotion of cell proliferation and migration by activating the Akt/mTOR signaling pathway. These results indicated that the missense variant *PPP4R3A* Asp409Asn is a candidate susceptibility gene for FNMTC, providing new insights into the diagnosis and intervention of FNMTC.

## 1. Introduction

In recent years, the prevalence of thyroid nodules has reached 70% [1]. Various factors, including age, radiation exposure, and family history, contribute to the diagnosis of thyroid cancer in 10% to 15% of thyroid nodules [2]. The majority of thyroid cancers are non-medullary thyroid cancer (NMTC) arising from follicular cells, with 5% to 15% occurring in a familial form known as familial non-medullary thyroid cancer (FNMTC) [3]. FNMTC is defined as the presence of follicular epithelium-derived thyroid cancer in two or more first-degree relatives within a family, excluding other familial syndromes and environmental factors [4]. Studies have shown that FNMTC exhibits more aggressive behavior compared with sporadic cases, with higher rates of multifocality, bilateral lesions, extrathyroidal invasion, and lymph node metastasis. FNMTC also demonstrates earlier onset and higher recurrence rates [5,6,7]. Furthermore, individuals with a first-degree relative affected by thyroid cancer have a 5 to 9-fold higher risk of developing thyroid cancer, with a higher probability of familial syndrome occurrence when multiple family members are affected [8].

Although a small portion (approximately 5%) of FNMTC cases are associated with families predisposed to multiple organ tumor diseases, the majority (approximately 95%) of FNMTC cases occur independently. FNMTC exhibits characteristics such as low penetrance, racial differences, and a complex genetic background involving mutations at multiple chromosomal loci [9]. Several low-penetrance susceptibility gene variations associated with FNMTC have been identified, including *FOXE1* [10], *HABP2* [11,12], *C14orf93 (RTFC)* [13], *DUOX2* [14], *MAP2K5* [15], *NOP53* [16], *CHEK2* [17], *SPRY4* [18], *POT1* [19], *WDR77* [20], and *CTSF* [21]. Candidate chromosomal susceptibility loci have also been proposed, such as *NMTC1* (2q21) [22,23], *MNG1* (14q32) [23], *TCO* (19p13.2) [24], *fPTC/PRN* (1q21) [23], *FTEN* (8p23.1–p22) [25], 6q22 [24], 4q32 [26], and 8q24 [27]. However, most of these variations are specific to particular families and lack validation in other families or replication in different populations. Consequently, FNMTC exhibits significant genetic heterogeneity, and the precise susceptibility genes remain unclear.

Protein phosphatase 4 regulatory subunit 3A (*PPP4R3A*), forms a highly conserved trimeric complex called protein phosphatase 4 (PP4) with PPP4C and PPP4R2. This complex plays a role in regulating the cellular processes, including DNA damage repair [28,29]. *PPP4R3A* is widely expressed in various tissues and organs, participating in multiple cellular functions, such as cell proliferation, apoptosis, and cell cycle regulation [29,30]. Recent evidence suggests that *PPP4R3A* is also involved in tumorigenesis. It has been shown to exert anti-angiogenic effects in ovarian tumors by inhibiting the VEGFR-2-mediated PI3K/Akt/eNOS signaling pathway [31] and interacting with the tumor suppressor BLU, resulting in pro-apoptotic activity in cervical and ovarian cancers [32]. Conversely, a recent study suggested that *PPP4R3A* promotes carcinogenesis in lung adenocarcinoma by activating the Wnt/β-catenin signaling pathway [33].

In this study, we conducted whole-exome sequencing (WES) on a FNMTC pedigree consisting of four affected members across two generations, all diagnosed with papillary thyroid carcinoma (PTC). Through WES analysis, we identified a novel variant (NC_000014.8:g.91942196C>T) in the *PPP4R3A* gene that exhibited a pattern of genetic segregation with the phenotype within this pedigree. Subsequently, we performed functional studies to investigate the impact of *PPP4R3A* and its missense variant Asp409Asn in thyroid cancer. Our findings suggest that the missense variant *PPP4R3A* Asp409Asn is a candidate susceptibility gene for FNMTC.

## 2. Materials and Methods

### 2.1. Ethics Statement and Subjects

The study included a FNMTC kindred consisting of four affected individuals. Figure 1 provides an overview of the family tree, illustrating the presence of papillary thyroid cancer across two generations. At total of 248 unrelated Chinese cases, with apparently sporadic NMTC (without any positive family history for the disease), were also included in this study.

This study was approved by and performed in accordance with the Ethics Committee of the Union Hospital Tongji Medical College of Huazhong University of Science and Technology (Approval number: 2017-S062).

### 2.2. DNA Extraction, Whole Exomes Sequencing, and Sanger Sequencing

Genomic DNA was extracted from peripheral blood samples of affected and unaffected individuals using TIANamp Genomic DNA Kit (DP304-03, TIANGEN Biotech, Beijing, China). The concentration of the DNA samples was measured using the Qubit 3.0 Fluorometer (Thermo Fisher Scientific, Waltham, MA, USA) in combination with the Qubit^TM^ dsDNA HS Assay Kit (Q32854, Thermo Fisher Scientific). Agarose gel electrophoresis was then used as part of quality control to confirm the integrity of all DNA samples, evaluating the presence of intact DNA fragments, the degree of DNA degradation, and potential RNA contamination.

Whole-exome sequencing (WES) was performed in six family members (II 2, II 3, III 3, III 5, III 7, and III 9, marked with * in Figure 1) from the kindred. Genomic DNA was fragmented into 180bp fragments, enriched for exons using the Agilent SureSelect HumanExomeV5 (50 Mb) exon capture kit (Agilent, Santa Clara, CA, USA), and subjected to library preparation for Illumina sequencing, following the manufacturer’s instructions. Then, Sentieon (version 201808.05) was used for the detection of single nucleotide polymorphisms (SNPs) and insertion-deletion (InDel) variants. The bioinformatics analysis followed the steps of alignment and sorting, duplicate removal, local realignment, base quality score recalibration (BQSR), and variant quality score recalibration (VQSR) for quality control and variant filtering, resulting in the final variant results. Sequencing was performed on the Illumina HiSeq × 10 sequencing platform, using the PE 150 sequencing mode, ensuring a minimum sequencing depth of 100×, and a sequencing yield of Agilent SureSelect V5 ≥ 10G. This technology was provided by Mingma Biotechnology Co., Ltd. (Shanghai, China).

The variants identified by WES were validated in both the family and in a cohort of 248 patients with sporadic NMTC using Sanger sequencing (primers information is listed in Appendix A). Purified PCR products were sequenced using the BigDye 3.1 Terminator sequencing kit on an ABI3730xl sequencer (Applied Biosystems, Waltham, MA, USA). This technology was provided by Tianyi Huiyuan Biotechnology Co., Ltd. (Wuhan, China).

### 2.3. Filtering Criteria of Whole Exomes Sequencing Data and Validation

The intersection of filtered whole-exome data was obtained for four thyroid cancer patients (II 3, III 3, III 5, III 7) within the family. The analysis involved excluding the normal elderly control (II 2), removing intronic variants, eliminating synonymous mutations, and excluding fully homozygous genes. Variants were further filtered and analyzed using ExAC (https://gnomad.broadinstitute.org/downloads#exac-variants, accessed on 30 October 2022) and the 1000 Genomes Project (ftp://ftp.1000genomes.ebi.ac.uk/vol1/ftp/releas, accessed on 30 October 2022), with mutations having an Asian allele frequency greater than 0.001 being excluded. Pathogenicity prediction models, such as SIFT, Polyphen2, and Mutation Taster, were applied to assess the impact of the variant on the protein sequence, with a specific focus on genes associated with thyroid function and cancer: (1) SIFT (dbNSFP version 3.0): The SIFT_score reflects the SIFT score, with lower values indicating a higher probability of causing alterations in protein structure or function. In cases where the variant affects multiple protein sequences, the minimum SIFT value is selected. SIFT_pred represents the prediction outcome, with T denoting “Tolerated” and D denoting “Deleterious”. A D prediction indicates a deleterious effect, where the score is ≤0.05. (2) Polyphen2_HDIV (dbNSFP version 3.0): It utilizes PolyPhen2 based on the HumanDiv database to predict the impact of mutations. Polyphen2_HDIV_score represents the PolyPhen2 score, with higher values indicating a greater likelihood of causing changes in protein structure or function. Polyphen2_HDIV_pred provides the prediction outcome, categorized as follows: D: “Probably damaging” (Polyphen2_HDIV_pred ≥ 0.957), P: “Possibly damaging” (0.453 ≤ Polyphen2_HDIV_pred ≤ 0.956), or B: “Benign” (Polyphen2_HDIV_pred ≤ 0.452). (3) MutationTaster (dbNSFP version 3.0): MutationTaster_score represents the MutationTaster score, where a higher value indicates a more reliable prediction result. MutationTaster_pred is the prediction result, with values of A: “Disease-causing automatic”, D: “Disease-causing”, N: “Polymorphism”, or P: “Polymorphism automatic”. Both A and D indicate potentially harmful variants at the given site. A preliminary screening of candidate genes and their corresponding loci was performed.

### 2.4. Cell Culture

Three cell lines were used in the study. Nthy-ori 3-1 (human normal thyroid epithelial cell line) was obtained from the European Collection of Authenticated Cell Cultures (ECACC). TPC-1 and B-CPAP (human papillary thyroid cancer cell line) were obtained from the American Type Culture Collection (ATCC).

Nthy-ori 3-1 and B-CPAP cells were cultured in the RPMI-1640 medium (Biological Industries, Kibbutz Beit Haemek, Israel) supplemented with 10% fetal bovine serum (FBS; Biological Industries, Israel). TPC-1 cells were cultured in DMEM medium (Biological Industries, Israel) with 10% FBS. All cells were maintained in a standard humidified incubator at 37 °C under a 5% CO_2_ atmosphere.

### 2.5. Lentivirus Preparation and Construction of Cell Lines

To stably knockdown *PPP4R3A* in Nthy-ori 3–1, TPC-1, and B-CPAP cells, we obtained plasmids containing short hairpin RNAs (shRNAs) targeting *PPP4R3A* from General Biol (Chuzhou, China). The specific targeting sequences of the shRNAs can be found in Appendix A. Lentiviruses were generated by co-transfection of the above constructs with packaging plasmids (pMD2.G and psPAX2) into HEK293T cells using Opti-MEM (Gibco, Thermo Fisher Scientific, Waltham, MA, USA) and Lipo8000^TM^ Transfection Reagent (Beyotime Biotechnology, Shanghai, China). After centrifugation at 3000× *g* for 15 min, the HEK293T medium containing lentiviruses carrying *PPP4R3A* shRNA sequences and control sequences were collected through 0.45 μm pore size filters.

The lentiviruses expressing wild-type *PPP4R3A* gene (NM_001366432.2) and mutant *PPP4R3A* gene with g.91942196C>T mutation (referred to as *PPP4R3A*: g.91942196C>T in the following text) were directly purchased from General Biol (Chuzhou, China). The negative empty vector (without wild type or mutant sequence) for pLVX-puro was used as a control for the experiments. Subsequently, the lentiviruses were added to each cell line in the presence of 10 µg/mL Polybrene (General Biol). A final concentration of 10 μg/mL Puromycin (Solarbio Science & Technology, Beijing, China) was used to select cell lines stably infected with the virus. Reverse transcription quantitative real-time PCR (RT-qPCR) and Western blot assays were used to confirm *PPP4R3A* downregulation and overexpression (detailed information provided in Section 2.8 and Section 2.9 of the “Materials and Methods” section).

### 2.6. Clonogenicity and Cell Proliferation Assays

In the colony forming assay, cells were seeded in 6-well plates at a density of 800 cells per well, and incubated at 37 °C for 7–10 days. Afterward, cells were fixed with 4% paraformaldehyde for 20 min and stained with 0.05% crystal violet for 30 min before being photographed.

For cell proliferation assay, a total of 3 × 10^3^ cells were seeded into 96-well plates and cultured for specific time intervals: 0 h, 24 h, 48 h, 72 h, and 96 h. Cell Counting Kit-8 (CCK-8, Selleck Chemicals, Houston, TX, USA) assay was then performed according to the manufacturer’s instructions. The cells were subsequently incubated for 1 h, and the absorbance at 450 nm wavelength of each well was measured using a microplate reader (Thermo Fisher, Waltham, MA, USA).

### 2.7. Cell Migration and Invasion Assays

Both the wound healing assay and transwell migration assay were employed to study cell migration. In the wound healing assay, cells were seeded into 6-well plates and incubated at 37 °C until they reached at least 90% confluence. A 200 μL pipette tip was then used to create a straight scratch across the cell monolayer, followed by the replacement of the medium with 2% FBS-containing medium. The scratches at the same location were observed under a microscope (Olympus, Tokyo, Japan) at 0 h, 12 h, and 24 h.

For transwell migration assay, a total of 2 × 10^5^ cells in 200 μL serum-free medium were plated to the top chamber of transwell plates (24-well insert, 8 μm pore size; Corning Costar, Cambridge, MA, USA). The bottom chambers were filled with 600 μL medium containing 20% FBS. After 24 h of incubation, cells were fixed with 4% paraformaldehyde for 20 min and stained with crystal violet for 30 min. Subsequently, the stained cells in randomly selected visual fields were counted and imaged using a microscope (Olympus).

Cell invasion ability was evaluated using transwell invasion assay. Transwell inserts were pre-coated with Matrigel (1:8 dilution; BD Biosciences, San Jose, CA, USA) before the cells were plated in the top chamber. The remaining steps of the protocol were identical to the transwell migration assay described above.

### 2.8. Quantitative Real-Time Polymerase Chain Reaction (qRT-PCR)

Total cellular RNA was extracted using TRIzol reagent (Vazyme, Nanjing, China) following the manufacturer’s instructions. The RNA concentration was measured using NanoDrop 2000C spectrophotometer (Thermo Fisher Scientific, Waltham, MA, USA), and then 1μg of total RNA was reverse transcribed into complementary DNA (cDNA) using HiScript III RT SuperMix for qPCR (Vazyme) according to the manufacturer’s instructions. Subsequently, the AceQ qPCR SYBR Green Master Mix (Vazyme) was used to perform the qRT-PCR reaction on a Bio-Rad CFX96^TM^ system (Bio-Rad Laboratories, Hercules, CA, USA). The relative expression levels of target genes were calculated using the 2^−ΔΔCT^ method, with normalization to the expression of the internal reference gene GAPDH in each sample. The primer sequences used in qRT-PCR are provided in Appendix A.

### 2.9. Western Blot

Total cellular proteins were extracted using ice-cold RIPA lysis buffer (Biosharp, Hefei, China) supplemented with protease inhibitor cocktail (MedChem Express; Monmouth Junction, NJ, USA), phosphatase inhibitor cocktail (MedChem Express) and Phenylmethylsulfonyl fluoride (PMSF; Solarbio Science & Technology). The protein concentration was determined using a BCA assay kit (Vazyme) following the manufacturer’s instructions. Protein samples were resolved on a 10% sodium dodecyl sulfate-polyacrylamide gel (SDS-PAGE) and transferred to a 0.45 μm nitrocellulose filter membrane (Merck Millipore, Billerica, MA, USA). The membrane was then blocked with 5% skim milk at room temperature for 1 h and subsequently incubated with primary antibodies at 4 °C overnight. The following antibodies were tested: anti-PPP4R3A (1:1000, A8500), anti-Phospho-Akt-T308 (1:500, AP1259), anti-Cyclin D1 (1:1000, A19038), anti-Bcl-2 (1:1000, A19693), anti-CDKN1A/p21 (1:1000, A19094), anti-Bax (1:1000, A18642) antibodies (Abclonal, Wuhan, China), anti-Akt (1:1000, #4691), anti-Phospho-Akt (Ser473) (1:800, #4060), anti-mTOR (1:1000, #2983), anti-Phospho-mTOR (1:1000, #5536T), anti-p70 S6 Kinase (1:1000, #9202), anti- Phospho-p70 S6 Kinase (1:1000, #9204), anti-4E-BP1 (1:1000, #9644), anti-Phospho-4E-BP1 (1:1000, #9451), anti-CDK4 (1:1000, #12790), anti-CDK6 (1:1000, #13331), anti-p53 (1:1000, #2524) antibodies (CST, Danvers, MA, USA), and anti-GAPDH (1:10,000, 60004-1-Ig, proteintech, Wuhan, China). The secondary antibodies were anti-mouse (#7076) or anti-rabbit (#7074) antibodies and were conjugated to horseradish peroxidase (HRP) (1:3000, CST). Finally, the bands were visualized using enhanced ECL reagents (Biosharp) on a ChemiDoc XRS+ chemiluminescent imaging system (Bio-Rad Laboratories).

### 2.10. Transcriptome Sequencing (RNA-seq)

For RNA-seq analysis, 5 × 10^6^ cells per group of TPC-1-sh-NC and TPC-1-sh-2 cells were submitted to Novogene (Beijing, China) for sequencing. Total cellular RNA was used as input material for RNA sample preparation. The library fragments were size-selected using the AMPure XP system (Beckman Coulter, Beverly, MA, USA) to obtain cDNA fragments ranging from 370 to 420 bp. After cluster generation, the prepared libraries were sequenced by the Illumina Novaseq 6000 platform, generating paired-end reads of 150 bp. The reads were then counted and mapped to each gene using FeatureCounts (v1.5.0-p3). Differential expression analysis between the two groups was then performed using the DESeq2 R package (1.20.0). Significantly differentially expressed genes were identified using the criteria of padj ≤ 0.05 and |log2(foldchange)| ≥ 1. To gain insights into the functional annotation of the differentially expressed genes, Gene Ontology (GO) enrichment analysis was conducted using the clusterProfiler R package (version 4.7.1, obtained from Bioconductor 3.17). Additionally, a pathway analysis was performed also using the clusterProfiler R package with a significance threshold of *p* value < 0.05, based on the Kyoto Encyclopedia of Genes and Genomes (KEGG) database, to identify the significant pathways associated with the differentially expressed genes.

### 2.11. Statistical Analysis

Statistical analysis was conducted using SPSS statistical software, version 26.0 (IBM Corp, Armonk, NY, USA) and GraphPad Prism, version 8.0 (GraphPad Software, San Diego, CA, USA). The results are presented as means ± standard deviation (mean ± SD). The differences between the two groups were evaluated by Student’s *t*-test. For comparisons involving multiple groups, one-way analysis of variance (ANOVA) was performed. *p* value < 0.05 was considered statistically significant.

## 3. Results

### 3.1. Clinical Characteristics of the NMTC Family

A kindred with four FNMTC patients came to our attention in the Union Hospital Tongji Medical College of Huazhong University of Science and Technology (Figure 1). To ensure consistent and standardized evaluation, we conducted a comprehensive thyroid-related physical examination and assessed the tumor markers for the included family members. Additionally, their medical histories were thoroughly investigated, and no other primary cancers were detected among the FNMTC patients or unaffected members of the family. Table 1 presents a summary of the clinical and pathological characteristics of these individuals. The four patients (II 3, III 3, III 5, III 7) diagnosed with PTC in this family displayed characteristics of bilateral and multifocal tumors. Among them, three patients (III 3, III 5, III 7) exhibited capsular invasion, and patients III 3 and III 5 additionally presented with intraglandular dissemination. All four PTC patients underwent total thyroidectomy and central or even lateral neck dissection. Subsequent follow-up assessments were conducted for these PTC patients. Notably, the average age of diagnosis for these individuals was at 28.5 (19–33) years. During the follow-up period, patient III 5, diagnosed with thyroid cancer at the age of 19, experienced a recurrence and lymph node metastasis, confirmed through aspiration, and subsequently underwent a re-operative surgery. All four PTC patients in the family received multiple radioactive-iodine therapies. Following postoperative TSH inhibition therapy, three patients (II 3, III 3, III 5) exhibited structural abnormalities, while one patient (III 7) presented with biochemical abnormalities.

### 3.2. Identification of PPP4R3A as a Candidate Susceptibility Gene through WES Analysis

Based on stringent criteria and filtration procedure, we obtained the intersection of whole-exome data (filtered data) from four thyroid cancer patients within the family (II 3, III 3, III 5, III 7). We excluded the elderly control individual II 2 (who had thyroid follicular cysts, considered a physiological change and thus used as a normal control). Subsequently, we integrated the unique SNPs and InDels specific to the patients within the family. To ensure the expression and pathogenic functionality of the variants, we excluded intronic and synonymous mutations, retaining the non-synonymous mutations in the exonic and splice site regions. We also eliminated fully homozygous genes with identical alleles at the same positions on homologous chromosomes. As a result, we preliminarily identified 12 candidate genes and conducted further screening based on allele frequencies and pathogenicity prediction scores. The summary can be found in Table 2.

Considering the overall incidence rate of thyroid cancer is approximately 11.95 × 10^−5^ among different countries [34], with FNMTC accounting for about 5% of NMTC, we excluded variants (*DUSP16*, *CHD4*, *SSPO*, *NLRP9*, *ANO2*, *CTBS*, *OR51B4*) from the general population with allele frequencies higher than 0.001 based on the 1000 Genomes Project and ExAC databases. According to the guidelines established by the American College of Medical Genetics and Genomics/Association for Molecular Pathology (ACMG/AMP) for standardized assessment of pathogenicity in Mendelian disease variants [35], the presence of this variant within the current pedigree, with a likelihood of 12.5% (1/2)^3^, can only be considered “supporting evidence” of its pathogenicity. Therefore, a comprehensive evaluation of the variant’s pathogenicity requires its investigation in healthy populations and individuals with sporadic thyroid cancer, modeling analysis to elucidate its impact on protein function, and the execution of in vitro functional experiments. Subsequently, we applied three pathogenicity prediction models (SIFT, Polyphen, and MutationTaster) to assess the impact of mutations in the remaining five candidate genes: *PPP4R3A*, *MANSC1*, *IQSEC3*, *MYL1*, and *VWF*. The corresponding scores were 2, 1, 3, 2.5, and 3, respectively. Finally, we were particularly intrigued by the gene *PPP4R3A*, which has shown high pathogenicity prediction scores and has been reported in various malignancies in recent years but has not been studied in thyroid cancer before. Hence, we selected *PPP4R3A*: g.91942196C>T as our candidate susceptibility gene for further investigation in this family.

To validate whether the candidate gene *PPP4R3A* Asp409Asn mutation conforms to the basic inheritance pattern of FNMTC, we performed validations within the family, between families, in sporadic thyroid cancer cases, and the healthy population, following the gene mapping strategy outlined in Figure 2A. For this particular family, we conducted intrafamilial validation on members I 4, II 4, and III 9. Genomic DNA fragments of I 4 and II 4 were amplified through PCR and validated using Sanger sequencing, while WES data was directly searched for III 9. Sanger sequencing results revealed that both I 4 and II 4 were wild type (Figure 2B). A comprehensive search for the *PPP4R3A* Asp409Asn mutation in the WES data of member III 9 yielded no results. Thus, we concluded that mutant *PPP4R3A*: g.91942196C>T segregates with the gene and phenotype in this family (Figure 1).

For validation in sporadic non-medullary thyroid carcinoma (SNMTC) populations, we referred to The Cancer Genome Atlas (TCGA) database (https://www.cancer.gov/about-nci/organization/ccg/research/structural-genomics/tcga, accessed on 22 December 2022). In this study, we downloaded a dataset of 490 cases with valid SNP data for thyroid carcinoma from the TCGA database. Using the R package maftools (version 2.14.0, obtained from Bioconductor 3.17) in R language, we analyzed and searched for the *PPP4R3A* Asp409Asn mutation. Among the sporadic NMTC patients included in the TCGA database, the *PPP4R3A* gene mutation Asp409Asn was absent in all 490 cases. Additionally, we collected peripheral blood samples from 248 patients with SNMTC at our institution. Genomic DNA was extracted from the samples, and PCR amplification followed by Sanger sequencing was performed on the genomic DNA fragments of these patients. The results showed that the *PPP4R3A:* g.91942196C>T mutation was absent in all 248 cases (Figure 2B).

Taken together, these findings indicate that the *PPP4R3A*: g.91942196C>T mutation is not commonly observed in SNMTC patients, suggesting that hereditary and sporadic NMTC patients do not share the same pathogenic mechanism. Furthermore, to understand the prevalence of the *PPP4R3A* Asp409Asn mutation in the general population, we confirmed its absence in the East Asian population by referencing the Genome Aggregation Database (gnomAD) (https://gnomad.broadinstitute.org/, accessed on 22 January 2023). It should be noted that although this database includes samples from the general population, the prevalence of thyroid cancer is only about 0.012%, allowing us to approximate the general population as a healthy population. In addition, we obtained WES data from eight thyroid cancer patients in two other FNMTC families. Unfortunately, none of these eight patients exhibited the *PPP4R3A* mutation (Appendix A).

We used the UniProt website (https://www.uniprot.org/, accessed on 18 May 2023) to query and align the amino acid sequences of the *PPP4R3A* gene in eight species. The amino acid residue at position 409 of the *PPP4R3A* sequence has remained aspartic acid, displaying high conservation throughout species evolution, indicating its potential significance within the PPP4R3A protein (Figure 2C). Additionally, leveraging the protein crystal structure model of PPP4R3A available on UniProt (https://www.uniprot.org/uniprotkb/Q6IN85/entry, accessed on 28 August 2023), we used PyMOL software (version 2.5.7) to generate protein models illustrating both the native wild-type structure and the Asp409 mutated to asparagine (Asn) at the specific amino acid site (Figure 2D). Our homology modeling analysis suggests that the Asp409Asn mutation in *PPP4R3A* could exert an influence on its functionality by potentially inducing steric hindrance, thereby impacting protein folding and stability.

### 3.3. Knockdown of PPP4R3A Promotes Cell Proliferation, Migration, and Invasion

Initially, we conducted knockdown experiments targeting *PPP4R3A* to preliminarily validate its role as a tumor-suppressor gene or oncogene in thyroid cancer. Three different shRNAs were transfected into the normal thyroid cell line Nthy-ori 3–1, as well as two PTC cell lines, TPC-1 and B-CPAP. Among them, sh-2 demonstrated effective knockdown of *PPP4R3A* at the mRNA level in all three cell lines (Figure 3A). Therefore, sh-2 was selected for subsequent experiments in this study. The established sh-2 knockdown cell lines, along with their negative control group (sh-NC), were then subjected to protein-level validation using Western blot analysis (Figure 3B,C).

Subsequently, we explored the effects of wild-type *PPP4R3A* knockdown on cell proliferation using CCK8 assay and colony formation assay. Notably, *PPP4R3A* knockdown significantly increased cell proliferation and colony formation (Figure 3D,E). Additionally, we observed that knockdown of *PPP4R3A*, both in normal thyroid cell line Nthy-ori 3–1 and two PTC cell lines TPC-1 and B-CPAP, resulted in a significant enhancement in cell migration and invasion compared to the control group (Figure 3F,G). Collectively, these functional experimental results indicate that *PPP4R3A* exhibits tumor suppressor functions.

### 3.4. Functional Characterization of Wild-Type PPP4R3A and the p. Asp409Asn Variant

Next, we investigated the effects of overexpression of wild-type *PPP4R3A* (WT) and the p. Asp409Asn variant (MUT) in three cell lines. The overexpression was validated using qRT-PCR and Western blot analysis (Figure 4A–C). In the CCK8 and colony formation assays, overexpression of *PPP4R3A* significantly suppressed cell proliferation and colony formation in all three cell lines compared with the control vector (Figure 4D,E). Similarly, in the transwell assay, the overexpression of *PPP4R3A* significantly inhibited cell migration and invasion (Figure 4F), consistent with the results obtained from the gene knockdown experiments, further supporting the growth-inhibitory role of *PPP4R3A*. In contrast, the p. Asp409Asn variant of *PPP4R3A* promoted cell proliferation, colony formation, and cell migration compared to both the NC group and the WT group (Figure 4D–F). However, in terms of cell invasion, the MUT group exhibited higher invasive capability compared to the WT group, although no significant difference was observed between the NC group and the MUT group (Figure 4G). These findings further suggest that *PPP4R3A* inhibits tumor formation, while the Asp409Asn variant may lead to the loss of its tumor-suppressive function and even exhibit certain oncogenic effects.

### 3.5. Effect of PPP4R3A and the p. Asp409Asn Variant on Cell Cycle-Associated Protein Expression

The functional experiments above suggest that *PPP4R3A* may play a crucial role in thyroid cell growth. As the cell cycle is closely associated with cell proliferation, we further investigated the impact of *PPP4R3A* and its variant p. Asp409Asn on the expression of cell cycle-related proteins in thyroid cells using Western blot analysis. We observed that knocking down *PPP4R3A* resulted in a significant upregulation of Cyclin D1, CDK4, and CDK6, which are involved in the G1-to-S phase transition, while concurrently downregulating the expression of p53 and p21, key regulators of the cell cycle (Figure 5A,B). Conversely, the overexpression of wild-type *PPP4R3A* yielded opposite results (Figure 5C,D). Interestingly, the overexpression of the *PPP4R3A* p. Asp409Asn mutant significantly increased the expression levels of Cyclin D1, CDK4, and CDK6, along with a notable decrease in p53 and p21 expression compared to the NC and WT groups (Figure 5C,D). These findings suggest that *PPP4R3A* may exert a negative regulatory role in thyroid cell proliferation by suppressing the expression of cell cycle-related proteins. However, the p. Asp409Asn mutation in *PPP4R3A* promotes the expression of these proteins, thereby explaining its enhancing effect on cell proliferation observed in the functional experiments.

### 3.6. Regulation of Akt/mTOR Signaling Pathway by PPP4R3A and the p. Asp409Asn Mutant

To elucidate the mechanisms underlying the tumor-suppressive effect of *PPP4R3A* in thyroid cancer, we conducted transcriptome sequencing to identify the downstream pathways regulated by *PPP4R3A*. The knockdown of *PPP4R3A* in TPC-1 cells resulted in significant RNA-level upregulation of 568 genes and downregulation of 217 genes (Figure 6A). GO enrichment analysis revealed the involvement of these differentially expressed genes (DEGs) in various tumor-related processes, including cell adhesion, cell fate determination and commitment, calcium ion binding, and negative regulation of the Wnt signaling pathway (Figure 6B), highlighting the crucial role of *PPP4R3A* in thyroid cancer development. KEGG pathway analysis of the DEGs demonstrated a close association between *PPP4R3A* and cancer signaling pathways, particularly PI3K/Akt (Figure 6C). Considering the significance of the PI3K/Akt pathway in FNMTC progression and its potential relationship with invasiveness [36], we hypothesize that *PPP4R3A* regulates thyroid cancer development through this pathway.

Subsequently, we performed Western blot analysis to validate the impact of *PPP4R3A* on key molecules in the PI3K/Akt/mTOR signaling pathway. *PPP4R3A* knockdown led to a significant increase in the expression of phosphorylated Akt at Thr308 and Ser473 sites, as well as enhanced mTOR phosphorylation. Furthermore, the phosphorylation levels of downstream genes of the AKT/mTOR pathway, namely P70 SK6 and 4E-BP1, were significantly elevated (Figure 6D). Conversely, the overexpression of wild-type *PPP4R3A* yielded contrasting results (Figure 6E). Intriguingly, the overexpression of the *PPP4R3A* p. Asp409Asn mutant notably increased the phosphorylation levels of Akt, mTOR, P70 SK6, and 4E-BP1 compared to the NC and WT groups (Figure 6E). Our findings suggest that *PPP4R3A* inhibits thyroid cell growth through the Akt-mTOR-P70 S6K/4E-BP signaling pathway. However, the p.Asp409Asn mutation in *PPP4R3A* not only eliminates its tumor-suppressive function but also promotes the activation of the Akt-mTOR-S6K/4E-BP pathway, thereby enhancing cell proliferation.

## 4. Discussion

Family inheritance plays a prominent role in the development and progression of thyroid cancer, necessitating genetic testing for assessing the risk associated with thyroid nodules. However, the identification of susceptibility genes for patients at high risk in non-syndromic FNMTC remains limited, with diagnosis primarily relying on family history. In this study, we employed advanced tools for identifying novel cancer susceptibility genes in Mendelian diseases, namely whole-exome sequencing (WES). Five candidate susceptibility genes (*PPP4R3A*, *MANSC1*, *IQSEC3*, *MYL1*, and *VWF*) were selected through screening. By combining pathogenicity prediction of mutations and the functional research progress of each gene, we ultimately focused on a novel missense mutation in the *PPP4R3A* gene and confirmed its association with the pathogenicity of thyroid cancer through subsequent functional assays. Based on the pedigree analysis (Figure 1), the pathogenic inheritance in this family originated from the maternal lineage. The *PPP4R3A*: g.91942196C>T variant displayed complete segregation with the phenotype, as all PTC patients (II 3, III 3, III 5, III 7) carried this mutation, while unaffected individuals (II 2, III 9) and the paternal lineage (I 4, II 4) exhibited the wild-type allele. These candidate genes were highly conserved in sporadic thyroid cancer and healthy populations, indicating distinct pathogenic mechanisms between hereditary NMTC patients and sporadic cases.

Among the four individuals diagnosed with PTC in this family, bilateral and multifocal tumors were observed, with three patients displaying tumor invasion of the thyroid capsule and intraglandular dissemination, as indicated in Table 1. The average age of diagnosis for these individuals was relatively young. Moreover, follow-up assessments revealed lymph node re-metastasis, and structural or biochemical abnormalities in these patients, highlighting the more aggressive nature and poorer prognosis associated with FNMTC.

Protein phosphatase 4 (PP4) is a serine/threonine protein phosphatase that forms a trimeric holoenzyme complex consisting of a catalytic subunit and regulatory subunits. It plays a critical role in various cellular processes, including DNA double-strand break repair and DNA damage regulation [28,29]. In 2005, Gingras et al. identified a novel component, PPP4R3A, in the PP4 complex. They observed that yeast cells lacking the homologous gene *Psy2* showed increased sensitivity to cisplatin, suggesting the potential involvement of *PPP4R3A* in cellular tolerance to cisplatin [37]. Subsequent studies revealed that *Psy2* influenced cellular sensitivity to cisplatin and other drugs by participating in DNA damage repair [38]. In our study, we investigated for the first time the role of *PPP4R3A* in thyroid cancer progression and its underlying mechanisms. Knockdown of *PPP4R3A* in vitro enhanced the proliferation, migration, and invasion abilities of normal thyroid cells and PTC cells (Figure 3), while its overexpression had the opposite effects (Figure 4), suggesting *PPP4R3A* as an important suppressor of thyroid cancer progression and metastasis. Consistent with the classical Knudson’s two-hit model, tumor suppressor genes typically involve a germline mutation as the first hit, followed by a second hit comprising diverse genetic alterations, including loss of heterozygosity, a second somatic mutation, methylation changes, or somatic recombination leading to partial uniparental heterodisomy [39]. These events ultimately result in the complete loss or inactivation of the tumor suppressor gene’s function within affected cells. Unfortunately, the unavailability of tumor tissue samples from the study subjects precluded further investigation into the presence of gene alterations associated with the second hit. Additionally, the lack of protein level assessment in the subjects’ tissues hindered the evaluation of whether the Asp409Asn mutation could potentially disrupt PPP4R3A protein expression, thereby limiting further supportive evidence. Moreover, Western blot analysis demonstrated that *PPP4R3A* knockdown significantly upregulated the expression of Cyclin D1, CDK4, and CDK6 involved in the G1 to S phase transition, while downregulating key regulators of the cell cycle, p53 and p21. Conversely, their overexpression yielded opposite effects (Figure 5). These findings align with a previous study by Kim et al., reporting the additive effect of paclitaxel and *PPP4R3A* in inducing G1 or G2 phase arrest during the cell cycle progression in ovarian cancer [40]. Based on this, we speculate that *PPP4R3A* may inhibit thyroid cell proliferation by inducing G1 phase arrest.

Our further mechanistic investigations revealed that *PPP4R3A* effectively suppressed the Akt-mTOR-P70 S6K/4E-BP1 axis in thyroid cells, while the *PPP4R3A* Asp409Asn mutation activated this pathway (Figure 6). Conversely, the *PPP4R3A* Asp409Asn mutation not only abolished the negative regulatory functions of wild-type *PPP4R3A*, but also exhibited certain promoting effects on proliferation and migration by activating the Akt/mTOR pathway. Thus, we consider the *PPP4R3A* Asp409Asn mutation a loss-of-function mutation. The aberrant activation of the AKT/mTOR pathway is frequently observed in cancer, promoting unrestricted tumor growth through enhanced protein synthesis and cancer cell metabolism [41]. The PI3K/Akt/mTOR pathway is primarily activated at different targets in diseases with *BRAF* and *RAS* mutations, and these specific combinations have been shown to induce thyroid cancer progression in mouse models [42]. Zhu et al. identified three susceptibility genes potentially participating in the occurrence and development of FNMTC through the PI3K/Akt signaling pathway [36]. The accumulation of genetic alterations in the PI3K/Akt pathway may increase the invasiveness of thyroid cancer, potentially explaining the more aggressive nature of FNMTC compared to SNMTC. Given the crucial role of the PI3K/Akt/mTOR pathway in the occurrence and development of thyroid cancer, further investigation of the specific downstream mechanisms by which *PPP4R3A* regulates the activity of this axis is of great significance.

While our study strongly implicated the *PPP4R3A*: g.91942196C>T variant in FNMTC development, a replication of this variant in the two additional FNMTC pedigrees was not observed (Appendix A). While not ideal, it should be noted that such results are not uncommon in previous studies. One possible reason could be the limited sample size, which might not be sufficient to establish the distribution of this mutation in FNMTC. Future investigations should prioritize the collection of more pedigrees to reduce biases. Additionally, the commonly observed genetic heterogeneity in inherited diseases suggests that multiple gene variants can contribute to the same disease. In other words, a disease is not determined by a single factor. Existing studies have not yet identified important genetic loci that can be replicated in other pedigrees, suggesting that the genetic mechanisms underlying FNMTC may be complex. Reported susceptibility genes and loci, including *FOXE1* [10], *C14orf93 (RTFC)* [13], *DUOX2* [14], *MNG1* (14q32) [23], *TCO* (19p13.2) [24], *FTEN* (8p23.1–p22) [25], among others, have not been validated in pedigrees from other literature studies. Therefore, we proposed a hypothesis that susceptibility genes for FNMTC are family-specific pathogenic genes, and each family may possess unique pathogenic mechanisms that collectively contribute to the susceptibility profile of FNMTC. To comprehensively explore FNMTC pathogenesis, it is imperative to incorporate more FNMTC pedigrees to expand and refine the FNMTC gene pool.

In conclusion, our study provided initial evidence supporting the involvement of *PPP4R3A*: g.91942196C>T as a susceptibility gene in FNMTC. Further research is necessary to ascertain the presence of this variant in additional FNMTC pedigrees and elucidate the downstream mechanisms involving *PPP4R3A* in thyroid cancer. High-throughput gene sequencing technologies offer promising opportunities for screening pathogenic genes. Our findings contribute to the potential application of *PPP4R3A* as a molecular diagnostic marker in FNMTC, providing new insights into the diagnosis and intervention of FNMTC.

## Figures and Tables

**Figure 1 biomedicines-12-00244-f001:**
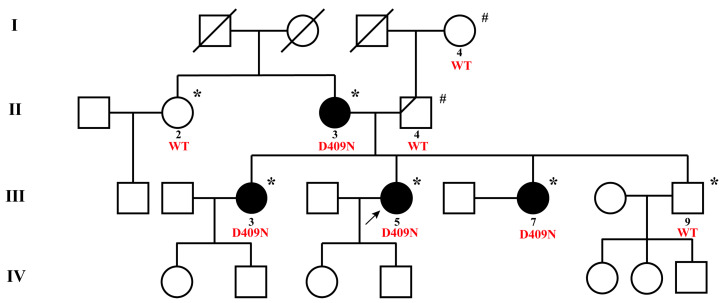
The pedigree of the FNMTC family. Squares represent males, and circles represent females. Filled circles and squares represent diagnosed PTC patients. Short diagonal lines represent patients with thyroid nodules, and long diagonal lines represent deceased members. The black arrow points to the FNMTC family proband. Subjects participating in this study are labeled with numbers. * denotes subjects analyzed by whole-exome sequencing; # denotes subjects analyzed by Sanger sequencing. “WT” stands for Wide Type, *PPP4R3A* wild type gene; “D409N” marks family members carrying the p.(Asp409Asn) mutation in the *PPP4R3A* gene.

**Figure 2 biomedicines-12-00244-f002:**
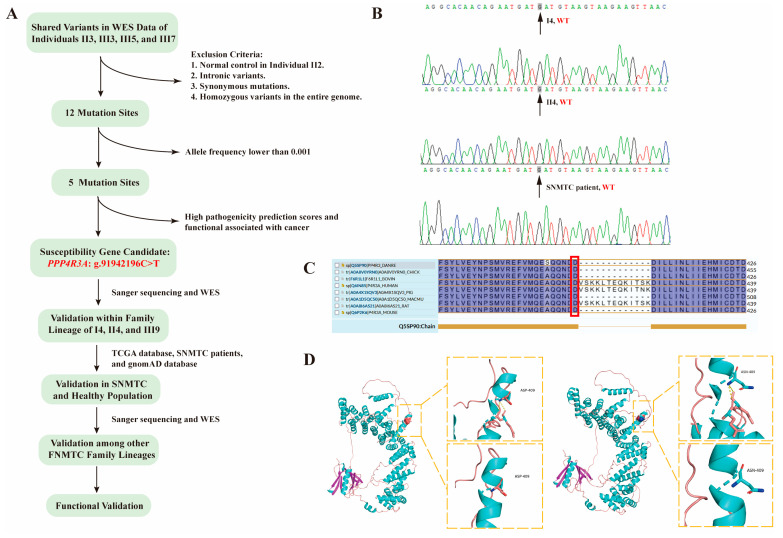
*PPP4R3A* was identified as a susceptibility gene in a FNMTC family through WES analysis. (**A**) Flowchart illustrating the susceptibility gene screening strategy; (**B**) Sanger sequencing results of family members I 4, II 4, and a representative SNMTC sample from our institution. The black arrow indicates the G to A substitution at the mutated locus of *PPP4R3A* (found in family members II 3, III 3, III 5, and III 7); (**C**) Conservation analysis of the p.Asp409 position across different species, with the red frame highlighting the amino acid aspartic acid at position 409. (**D**) The left panel depicts the crystal structure of the wild-type PPP4R3A protein, while the right panel displays the crystal structure of the PPP4R3A protein with the p.Asp409Asn mutation.

**Figure 3 biomedicines-12-00244-f003:**
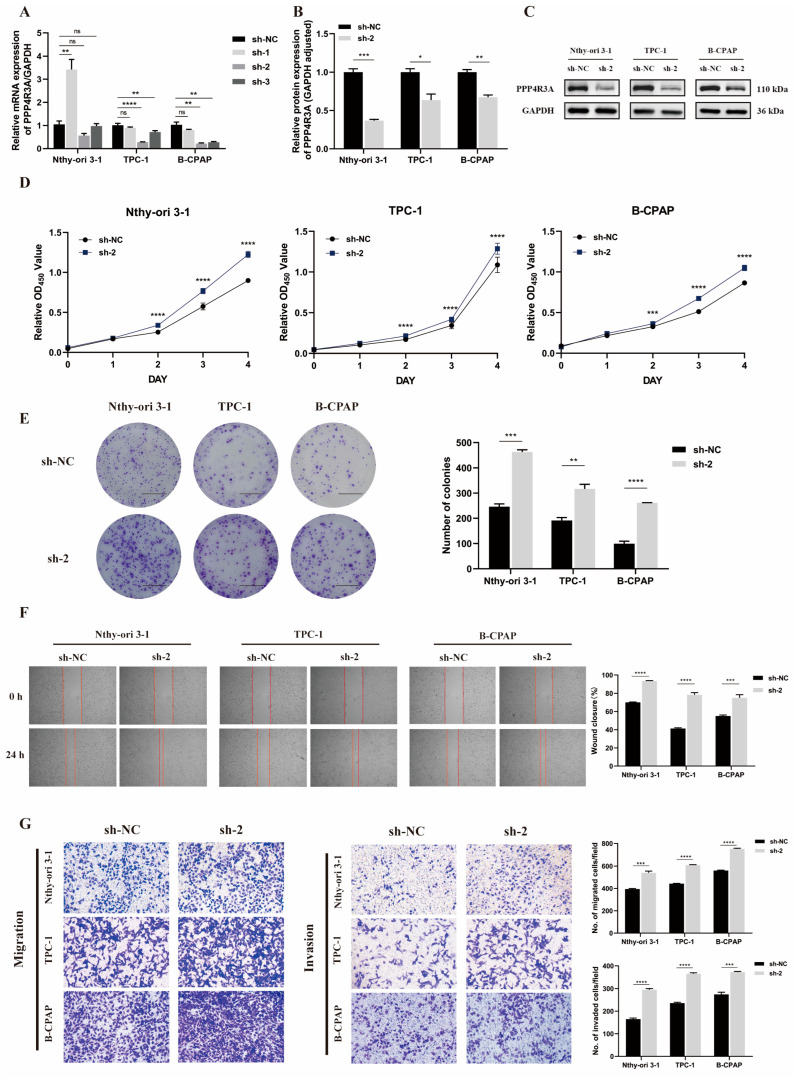
The effects of stable *PPP4R3A* knockdown in three thyroid cell lines (Nthy-ori 3–1, TPC-1, B-CPAP). (**A**) Quantitative RT-PCR analysis to validate the knockdown efficiency of *PPP4R3A* using shRNAs (sh-1, sh-2, and sh-3); (**B**,**C**) Validation of one specific shRNA (sh-2) targeting PPP4R3A protein expression in three different cell lines using Western blots. GAPDH was used as an internal control. Quantification data represent mean ± SD for three independent experiments. Appendix A provides duplicate results; (**D**) CCK8 assay measured the growth of Nthy-ori 3–1, TPC-1, and B-CPAP cells treated with the control shRNA(sh-NC) and sh*PPP4R3A* (sh-2), respectively, over a five-point time course; (**E**) Colony formation assay was carried out to evaluate the cellular colony-forming ability. The right panel shows quantification data. (Scale bars, 1 cm); (**F**) Wound healing assay to detect cell migration ability. The right panel presents quantification data; (**G**) Transwell migration and invasion assays to determine the migration and invasion ability of three thyroid cell lines, respectively (zoom 100×). The right panel provides quantification data. Error bars indicate standard deviation. Statistical significance is denoted as * *p* < 0.05, ** *p* < 0.01, *** *p* < 0.005, **** *p* < 0.001.

**Figure 4 biomedicines-12-00244-f004:**
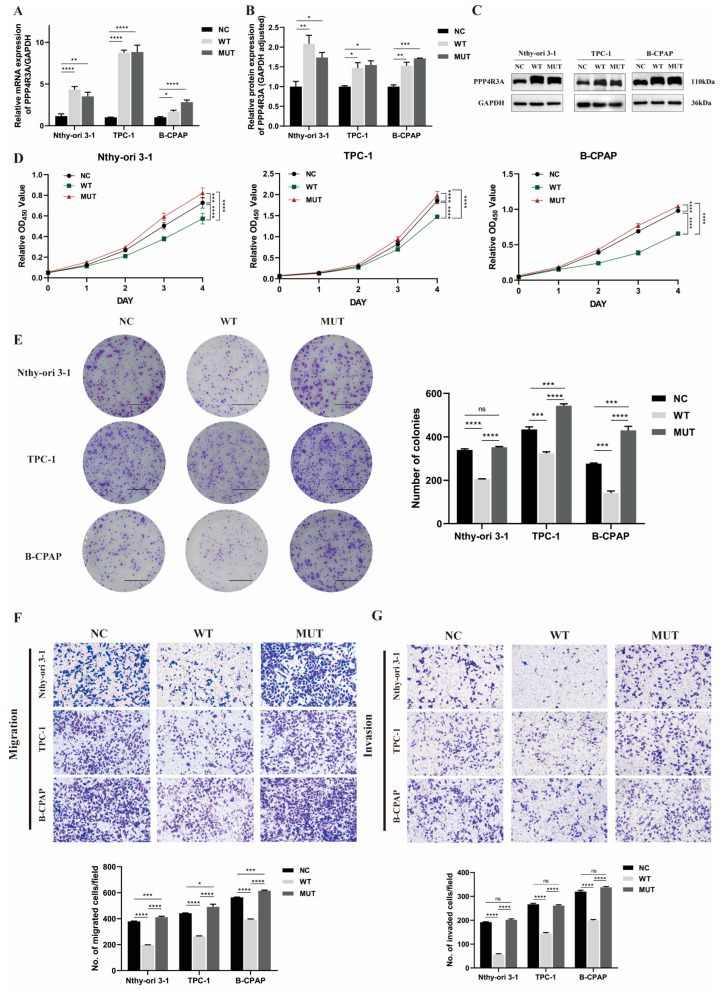
The effects of stable overexpression of wild-type *PPP4R3A* and the p. Asp409Asn variant in three thyroid cell lines (Nthy-ori 3-1, TPC-1, B-CPAP). (**A**) Quantitative RT-PCR analysis to validate the stable overexpression of wild-type (WT) and Asp409Asn mutant *PPP4R3A* (MUT) using lentivirus transfection in three different cell lines; (**B**,**C**) Western blot to detect protein overexpression of WT and MUT PPP4R3A, with GAPDH serving as an internal control. Quantification data represent mean ± SD. for three independent experiments. Appendix A provides duplicate results; (**D**) CCK8 assay to measure the growth of Nthy-ori 3–1, TPC-1, B-CPAP cells infected with empty vector (NC), WT, and MUT lentivirus, respectively, over a five-point time course; (**E**) Colony formation assay to evaluate the cellular colony-forming ability. The right panel shows quantification data. (Scale bars, 1 cm.); (**F**) Transwell migration assay to detect the migration ability of three thyroid cell lines (zoom 100×). The lower panel presents quantification data; (**G**) Transwell invasion assay to determine the invasion ability of three thyroid cell lines (zoom 100×), with the lower panel presenting quantification data. Error bars indicate standard deviation. Statistical significance is denoted as * *p* < 0.05, ** *p* < 0.01, *** *p* < 0.005, **** *p* < 0.001.

**Figure 5 biomedicines-12-00244-f005:**
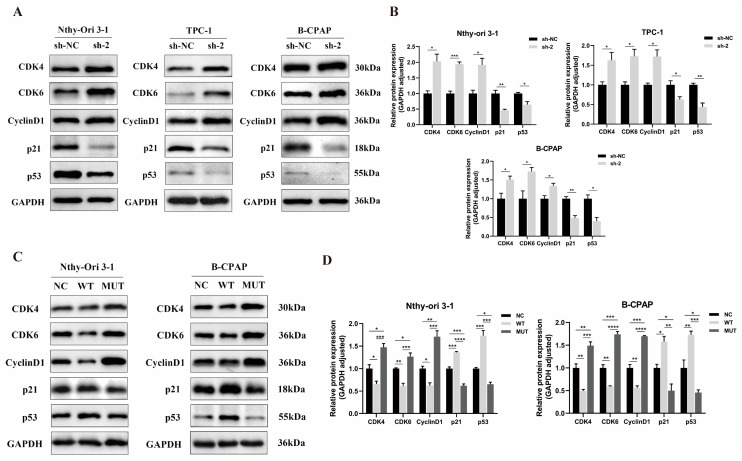
Impact of *PPP4R3A* and p. Asp409Asn variant on cell cycle-associated protein expression in thyroid cell lines. (**A**,**B**) Western blot analysis depicting the levels of cell cycle-related proteins in three thyroid cell lines (Nthy-ori 3-1, TPC-1, B-CPAP) treated with control shRNA (sh-NC) and sh*PPP4R3A* (sh-2). Quantification data are presented as mean ± SD for three independent experiments, normalized to the internal expression of GAPDH; (**C**,**D**) Western blot analysis illustrating the levels of cell cycle proteins in normal thyroid cell line Nthy-ori 3-1 and the PTC cell line B-CPAP overexpressing wild-type (WT) and mutant (MUT) *PPP4R3A*. Quantification data are shown as mean ± S.D. for three independent experiments. Appendix A provides duplicate results. Error bars represent standard deviation. * *p* < 0.05, ** *p* < 0.01, *** *p* < 0.005, **** *p* < 0.001.

**Figure 6 biomedicines-12-00244-f006:**
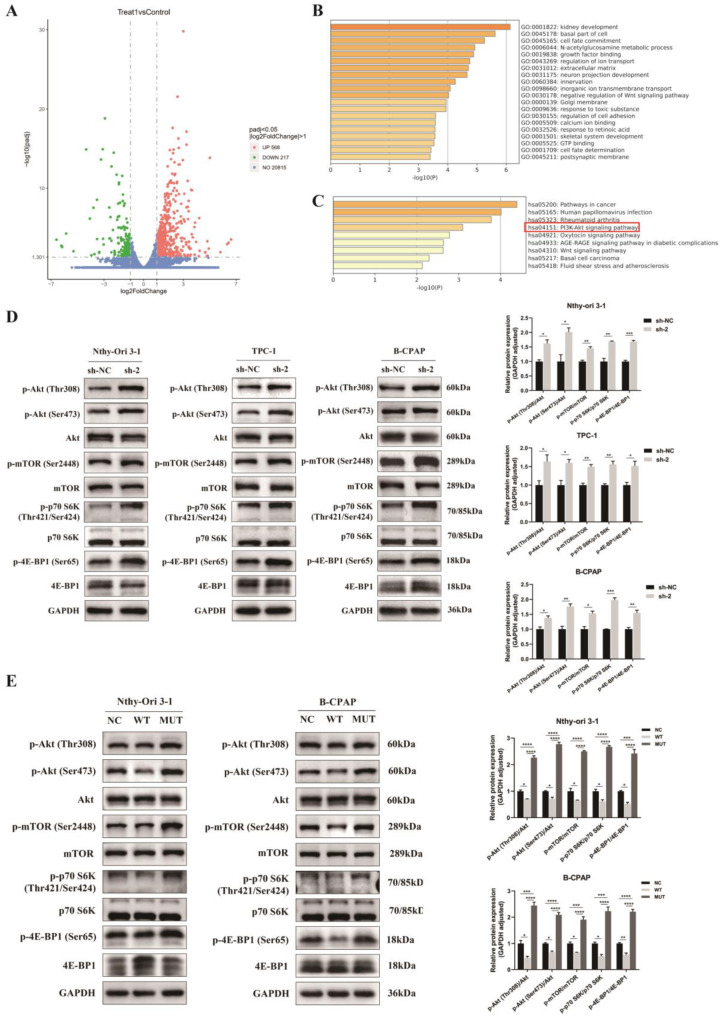
*PPP4R3A* and p.Asp409Asn mutant regulate the growth of thyroid cells via the Akt/mTOR signaling pathway. (**A**) Volcano plot representing differentially expressed genes (DEGs) identified through RNA-seq analysis in TPC-1 cells transfected with *PPP4R3A* sh-RNA 2 compared with sh-NC. The horizontal axis indicates the fold change (log2FoldChange) of gene expression between the sh-2 and sh-NC groups. The vertical axis represents the significance level of expression difference (−log10padj), with up-regulated genes denoted by red dots and down-regulated genes by green dots; (**B**,**C**) Enrichment analysis of DEGs after sh*PPP4R3A* transfection using Gene Ontology (GO) and Kyoto Encyclopedia of Genes and Genomes (KEGG) pathways. The PI3K/Akt signaling pathway is highlighted within the red box; (**D**) Western blot analysis confirming that *PPP4R3A* knockdown enhanced Akt/mTOR pathway activation in the normal thyroid cell line Nthy-ori 3–1 and two PTC cell lines (TPC-1 and B-CPAP). Quantification data are presented on the right panel; (**E**) Western blot analysis showing the differential effects of overexpressing wild-type (WT) and Asp409Asn mutant (MUT) *PPP4R3A* on Akt/mTOR pathway activation in the normal thyroid cell line Nthy-ori 3-1 and PTC cell line B-CPAP. Quantification data represent mean ± SD. for three independent experiments. Appendix A provides duplicate results. Error bars indicate standard deviation. * *p* < 0.05, ** *p* < 0.01, *** *p* < 0.005, **** *p* < 0.001.

**Table 1 biomedicines-12-00244-t001:** Clinical-pathologic and follow-up characteristics of the included family members.

Patient	I 4	II 2	II 3 *	II 4	III 3 *	III 5 *	III 7 *	III 9
Age (years)	87	60	58	59	38	36	34	32
Age at diagnosis (years)	NA	NA	33	NA	33	19	29	NA
Radiation exposure	N	N	N	N	N	N	N	N
Tumor history	N	N	Thyroid cancer	N	Thyroid cancer	Thyroid cancer	Thyroid cancer	N
Abnormal tumor markers	N	N	N	N	N	N	N	N
Thyroid dysfunction	hypothyroidism	N	PM	N	PM	PM	PM	N
Abnormal thyroid ultrasound	N	Physiologic cyst	PM	Thyroid nodule	PM	PM	PM	N
Surgical resection	NA	NA	TT + CLND	NA	TT + CLND + ULND	TT + CLND + BLND	TT + CLND	NA
Stage of cancer (TNM) ^a^	NA	NA	T2N0aM0	NA	T2N1bM0	T2N1bM0	T2N1aM0	NA
**Pathological characteristics**								
Histologic type of cancer								
The left lobe	NA	NA	PTMC	NA	PTMC	PTC	PTMC	NA
The right lobe	NA	NA	PTC	NA	PTC	PTC	PTMC	NA
No. of tumors	NA	NA	2	NA	3	2	3	NA
Size of tumors (cm)	NA	NA	0.8 × 0.6	NA	1.8 × 1.0	1.6 × 1.4	0.8 × 0.7	NA
	NA	NA	2.1 × 1.6	NA	0.8 × 0.6	1.2 × 1.0	0.6	NA
	NA	NA	NA	NA	0.2 × 0.1	NA	0.5	NA
Tumor invasion	NA	NA	NA	NA	Capsular invasion	Capsular invasion	Capsular invasion	NA
Intraglandular dissemination	NA	NA	N	NA	N	Y	Y	NA
Lymph node metastasis	NA	NA	0/5 (C)	NA	8/13 (C) 2/10(L)	2/2 (C) 5/30(L)	6/10 (C)	NA
**Follow-up**								
Evidence of recurrence	NA	NA	N	NA	N	Y	N	NA
Re-operative surgery	NA	NA	N	NA	N	Y	N	NA
No. of radioactive-iodine therapies	NA	NA	1	NA	2	2	2	NA
Results of TSH inhibition therapy ^b^	NA	NA	Structural abnormality	NA	Structural abnormality	Structural abnormality	Biochemical abnormality	NA

Abbreviations: N: No; Y: Yes; NA: Not available; PM: Postoperative manifestations; TT: Total thyroidectomy; CLND: Central lymph nodes dissection; ULND: Uni-lateral lymph nodes dissection; BLND: Bilateral lymph nodes dissection; PTC: Papillary thyroid carcinoma; PTMC: Papillary thyroid micro-carcinoma; C: Central; L: Lateral. * Four patients diagnosed with thyroid cancer (II 3, III 3, III 5, III 7) within the family are marked. ^a^ Staging was based on the tumor-node-metastasis (TNM) classification of the American Joint Committee on Cancer. ^b^ The results of TSH inhibition therapy are as follows: good response/biochemical abnormality/structural abnormality/uncertain response.

**Table 2 biomedicines-12-00244-t002:** Primary candidate genes and pathogenicity prediction scores.

Gene	ID	Position	Allele Change	Region	Chromosome Locus	1000G_ALL	1000G_EAS	ExAC_ALL	ExAC_EAS	SIFT_Score	SIFT_Pred	Polyphen2 HDIV_Score	Polyphen2 HDIV_Pred	Mutation Taster_Score	Mutation Taster_Pred	Hazard Score
*PPP4R3A*	rs1376958785	chr14:91942196	C>T	exonic	14q32.12	NA	NA	NA	NA	0.24	T	1	D	1	D	2
*MANSC1*	rs781617757	chr12:12483454	G>C	exonic	12p13.2	NA	NA	0	0.0001	0.016	D	0.025	B	1	N	1
*IQSEC3*	rs769019293	chr12:283790	C>T	exonic	12p13.33	NA	NA	0	0.0001	0.001	D	0.961	D	1	D	3
*MYL1*	rs570955358	chr2:211179708	G>A	exonic	2q34	0.0002	NA	0	0.0006	0.007	D	0.702	P	1	D	2.5
*VWF*	rs771423537	chr12:6153513	C>T	exonic	12p13.31	NA	NA	0	0.0003	0.017	D	0.999	D	1	D	3
*DUSP16*	rs3809199	chr12:12630669	C>T	exonic	12p13.2	0.013	0.005	0.026	0.013	NA	NA	0.005	B	1	N	1
*CHD4*	rs74790047	chr12:6711144	A>C	exonic	12p13.31	0.0074	0.034	0.013	0.0093	0.267	T	0	B	0.939	N	1
*SSPO*	rs118190970	chr7:149475053	C>T	exonic	7q36.1	0.004	0.018	0.001	0.012	NA	NA	0.001	B	NA	NA	0
*NLRP9*	rs138496520	chr19:56228104	C>T	exonic	19q13.42	0.0026	0.013	0.0013	0.018	0.54	T	0.059	B	1	N	1
*ANO2*	rs17788563	chr12:5853474	C>T	exonic	12p13.31	0.015	0.025	0.0059	0.024	0.973	T	0.285	B	1	D	1
*CTBS*	rs15911	chr1:85029077	C>T	exonic	1p22.3	0.15	0.03	0.2	0.032	0.109	T	0.489	P	0.017	P	1
*OR51B4*	rs78511352	chr11:5322776-	G>A	exonic	11p15.4	0.01	0.044	0.0049	0.041	0.012	D	0.185	B	1	N	1

Abbreviations: NA: Not available; 1000 G: 1000 Genomes Project; ExAC: Exome Aggregation Consortium; ALL: Overall Population; EAS: East Asian Population; SIFT, Polyphen2, Mutation Taster applications are described in detail in the Section 2.

## Data Availability

The original contributions presented in the study are included in the article/Appendix A, further inquiries can be directed to the corresponding authors.

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
