# Peer review of "Identification of a Novel Germline PPP4R3A Missense Mutation Asp409Asn on Familial Non-Medullary Thyroid Carcinoma"

_biomedicines, 2024, doi:10.3390/biomedicines12010244_

Round 1

Reviewer 1 Report

Comments and Suggestions for Authors

Manuscript ID:biomedicines-2663698

Authors conducted a comprehensive study of a family with four patients affected by papillary thyroid carcinoma. First, they used WES, selecting candidate genes and variants appropriately. Finally, they selected the PPP4R3A gene and the p.D409N variant. This variant cosegregates with the four affected members of the family.

They then analyzed if this gene and the variant itself could be involved in cell proliferation and migration. They evaluate cell migration, RNA levels, protein expression and transcriptome analysis. When they knockdown the gene, cell proliferation increased. Overexpression suppressed cell proliferation. All data they present support PPP4R3A gene as a tumor suppressor gene.

According to the authors, this variant could be involved in the development of NMTC in this family and perhaps, in others, suggesting the PPP4R3A as a susceptibility gene in FNMTC.

Methodology is correct and comprehensive. The manuscript is well written and scientifically correct.

However, manuscript presents some caveats:

-This variant has a probability of being present in all four individuals simply by chance, of 12.5% (1/2)3. To be considered likely-pathogenic, and according to the guidelines, this percentage must be around <5% (PMID: 27236918). This fact should be mentioned in the manuscript.

-Through the manuscript, they suggest this gene is a tumor suppressor gene. According to the Knudson’s model, they could reinforce this statement studying tumor samples, looking for the presence of loss of heterozygosity, a second punctual mutation in the gene, methylation or somatic recombination leading to partial uniparental heterodisomy. If they do not have tumor samples or do not have analyzed them, they should mention they have not explored this possibility.

Lines 582-586, authors propose: “Therefore, we proposed a hypothesis that susceptibility genes for FNMTC are family-specific pathogenic genes, and each family may possess unique pathogenic mechanisms that collectively contribute to the susceptibility profile of FNMTC”.

And in lines 589-591, authors propose: “Further research is necessary to ascertain the presence of this variant in additional FNMTC pedigrees”.

I think, there is incongruence between both proposals. If each family presents a specific gene, the presence of this or other PPP4R3A variants will not be found in other families, and therefore, this gene would not be ever validated. Please, clarify.

Note:

-Genes should be written in italics.

Reviewer 2 Report

Comments and Suggestions for Authors

Well-written, comprehensive with functional analysis, lacking detail with some experimental details.

-          Line 99-101: There are more than one dsDNA assay for Qubit, might be good to mention exactly which one was used. Not sure what was assessed with agarose gel QC; was it high molecular weight human genomic DNA band and visible degradation?

-          Line 106: Sentieon version number required. Manufacturer details missing, like HiSeq missing. No details on how library prep was performed, was it with one of the Illumina library prep kits? There are (again) many of these.

-          Line 121: not clear which platform was used for filtering as this can be done with Ensembl or even Excel.

-          Line 126: since ExAc browser is not available, it might be useful just to have a short info how the data was accessed and manipulated, i.e. downloaded and which tools were used. Same applies to 1000 Genomes.

-          Line 175-177: Details of RT-qPCR and western blot (should be title case) missing: enzymes and primer information completely missing, and data. If these are the same as in paragraph 2.8 / line 208 and lines 230-240, it should be mentioned. The results should be discussed and presented as stated in the MIQE guidelines (https://doi.org/10.1373/clinchem.2008.112797). The assay seems to be an in-house assay, and there are no details on PCR efficiency, for example, which can in worst case scenario reverse the seen result.

-          216-219: Using just one reference gene is not recommended as three should be used, although it gives a good way comparing to Western blots since the same gene product was used there. Instead of delta-delta-Ct methods, LinRegPCR might be more accurate: https://bmcbioinformatics.biomedcentral.com/articles/10.1186/s12859-021-04306-1

-          Line 255-257: This possibly means Bioconductor version 3.18? If so, correct details should be given (3.8.1 ->3.1.8)

-          Line 277: Table1, this table should be in landscape format or using abbreviations/coding as at present very difficult to ready do to the overflowing text to multiple lines, like “Postoperati / ve / manifestati / ons” on FOUR lines.

-          Line 278-288: The legend interrupts the table so the rest of the table on page 8 is left without headers. The legend should be AFTER the last row/column of the table

-          Line 312: Similar problems with Table 2 as with Table 1.

-          Line 342: Bit more detail on the selection would be beneficial as not crystal clear from the generic link and these resources change from time to time

-          Line 343: Maftools is part of Bioconductor, and the version number should be mentioned as well, as customary in scientific reporting

-          Line 351

-          Line 356: The given link directs to an illegal gambling site

-          Line 383: Relative expression bar charts should be in log-scale. If plotted linearily, 10x overexpression looks very different  than 10X underexpression, i.e. values 10 vs 0.1. In log-scale the chart will not have a bias on overexpression

-          Figures 3-6 all have the same, common problem when relative expression is plotted – these should be in log scale to avoid underpresentation of downregulation, i.e. the scale from 0-1 is nowhere near the same as from 1-10. This can be highlighted if you plot bars with values 0.1 and 10: they are miles different although should represent equal change in relative expression.

-           Throughout the document: western blot -> Western blot.

Comments on the Quality of English Language

-          Throughout the document: western blot -> Western blot.

Otherwise excellent

Reviewer 3 Report

Comments and Suggestions for Authors

Authors have performed an excellent study by investigating a family with multiple members affected by thyroid non-medullary carcinoma. All findings of their investigation support a pathogenic role of a new variant occurring in the PPP4R3A gene.

I have only minor comments and suggestions:

- it is not fully clear the way that Authors chose to identify candidate genes (12 in the Results section and 5 in the Discussion). Genes as ANO2 and VWF are not known to be involved in any pathway linked to carcinogenesis;

- the gene variant identified seems to exert a loss-of-function effect. Authors wrote sentences, as the following in the Results section (These findings
further suggest that PPP4R3A inhibits tumor formation, while the D409N variant may render it non-functional or even exhibit a certain oncogenic role.), that may be read as to have a gain-of-function role;

- in silico analysis of structural protein modelling may add valuable information on the effect of the gene variant;

- in Fig. 2b, the addiction of the electropherogram of a patient carrying the gene variant may add value to data found.

Reviewer 4 Report

Comments and Suggestions for Authors

By NGS analysis, the authors here identify a missense variants in PPP4R3A in a Chinese family with FNMT and they provide supporting evidence suggesting its role as susceptibility factor in thyroid carcinoma. The study is interesting but, in my opinion, the inability to replicate their results in other families or sporadic cases represent a serious limitation. In addition, there are many concerns that need to be addressed.

Major concerns:

1.     In the Abstract and Introduction, the authors state that whole genome sequencing has been carried out on the investigated Chinese family. However, in Material and Methods and in all the other section of the manuscript, they referred to whole exome sequencing. Authors have to clarify this point.

2.     Identifying PPP4R3A:c.1225G>A (p.D409N) as possible susceptibility factor, the authors also searched for the variant in a wide cohort of 490 sporadic NMTC cases. The p.D409N variant was absent but it is not clear to the reviewer why the authors did not consider the possibility that different and possibly pathogenic variants in PPP4R3A can be present in this cohort. Furtherly, they also reported that WES data were available from two additive FNMTC  families, in which the variant was absent. Did the authors only consider the p.D409N variant or did they also search for other PPP4R3A variants or variants in shared genes with their family (e.g., selected genes in Table 2)?

3.     The authors used  shRNAs to silence PPP4R3A in specific thyroid cell lines. However, PPP4R3 seems to be only variably downregulated in the different cell lines but not silenced. So, this point should be better clarified in the different section of the manuscript.

4.     The authors suggest that the identified p.D409N variant can act as a loss-of function variant. How this change can have this effect remain not explained. Did the authors consider to generate mutant at this position to be used to carry out the same functional experiments? Are 3D models available for the protein? How can this change impact on protein folding?

Minor concerns:

1.     Please, variants should be annotated according to the HGVS nomenclature and RefSeq you referred should be reported.

2.     Please, a legend should be added to Figure S1.

Line 529 “PP4” should be probably “PPP4”.

Round 2

Reviewer 4 Report

Comments and Suggestions for Authors

I thank the authors very much for having considered my suggestions, modifying the manuscript in accordance with them.